# Design of a High-Bandwidth Uniform Radiation Antenna for Wide-Field Imaging with Ensemble NV Color Centers in Diamond

**DOI:** 10.3390/mi13071007

**Published:** 2022-06-26

**Authors:** Zhiming Li, Zhonghao Li, Zhenrong Shi, Hao Zhang, Yanling Liang, Jun Tang

**Affiliations:** 1Key Laboratory of Instrument Science and Dynamic Testing Ministry of Education, North University of China, Taiyuan 030051, China; lizhiming_19941120@yeah.net (Z.L.); zhenrongshi@yeah.net (Z.S.); zhanghao_1996@yeah.net (H.Z.); yanling_liang@yeah.net (Y.L.); 2Key Lab of Quantum Sensing and Precision Measurement, Taiyuan 030051, China; 3Institute of Instrument and Electronics, North University of China, Taiyuan 030051, China

**Keywords:** uniformity, high-bandwidth, radiation antenna, ensemble NV color centers

## Abstract

Radiation with high-efficiency, large-bandwidth, and uniform magnetic field radiation antennas in a large field of view are the key to achieving high-precision wide-field imaging. This paper presents a hollow Ω-type antenna design for diamond nitrogen-vacancy (NV) ensemble color center imaging. The uniformity of the antenna reaches 94% in a 4.4 × 4.4 mm^2^ area. Compared with a straight copper antenna, the radiation efficiency of the proposed antenna is 71.8% higher, and the bandwidth is improved by 11.82 times, demonstrating the effectiveness of the hollow Ω-type antenna.

## 1. Introduction

The use of nitrogen-vacancy (NV) color centers in diamonds has a wide range of application prospects in the measurement of electric fields [1,2], temperature [3,4,5,6], magnetic fields [7,8,9,10,11,12,13,14,15], stress and strain detection [16,17], biological imaging [18,19], and medical imaging [20]. The NV color center has a long electron spin coherence time at room temperature, with an accurate readout [21,22,23,24,25] that has significantly promoted the development of solid-state electron spin in the field of quantum science precision and detection sensing. Among them, a common method for the precise detection and sensing of diamond NV color centers is optically detected magnetic resonance (ODMR). Initially, diamond NV color centers were used to measure magnetic field strength at a specific location [15]. A certain degree of imaging of the magnetic field and temperature distribution was gradually realized subsequently [26,27]. In particular, the wide-field imaging technology of diamond the NV ensemble color center can realize wide-field precision measurement. However, the microwave antenna uniformity and other factors reduce its measurement accuracy. A uniform microwave radiation antenna can effectively improve the accuracy of wide-field measurement.

A variety of microwave radiation structures have been designed for high precision detection with NV color centers. For diamond NV color center-related experiments, the magnetic field component in the electromagnetic field generated by the radiating antenna is particularly important. While a traditional single straight copper wire can be used in ODMR, it produces an inhomogeneous magnetic field [27], and it is not suitable for wide-field imaging. A microwave planar loop antenna was achieved by K. Sasaki, where the uniform area of the antenna on the planar loop was a diameter of 1 mm, and the bandwidth was only 400 MHz [28]. N. Zhang designed a square split ring resonator, where the microwave field uniformity in the effective imaging area of the resonator was 92.7%. However, its uniform area was only 1.3 × 1.3 mm^2^ [29]. E. R. Eisenach proposed a loop gap resonator, which provides a 96.8% uniform MW drive for NV ensembles in a 32 mm^2^ area, but the bandwidth is only 80 MHz [30]. In addition, K. Bayat designed a dual-loop dual-aperture antenna. Although the normalized standard deviation was less than 5% of the uniform microwave magnetic field distribution, its uniform area was only 0.95 × 1.2 mm^2^ [31]. Currently, the field scale of uniform microwave radiation antenna design is mainly concentrated in the order of 1 mm^2^, and the radiation bandwidth is limited to 400 MHz.

A hollow Ω-type antenna for wide-field imaging with a diamond NV ensemble color center is designed in this work. Based on the conditions of the NV color center ensemble imaging system, the uniformity, high bandwidth, and high radiation efficiency of the antenna are then verified, respectively.

## 2. Experimental System and Diamond

### 2.1. Experimental System

We briefly introduce the principle of imaging experiments here, while a more detailed outline can be obtained in Ref. [25].

The experimental system used in this work is shown in Figure 1a. The optical path system is mainly composed of a 532 nm laser (MGL-III-532-100 mW, Changchun New Industries Optoelectronics Technology, Changchun, China), a photoelectric detector (PD, Photodetector, APD130A, Thorlabs, Newton, NJ, USA), a CMOS camera (CS2100M-USB, Thorlabs, Newton, NJ, USA), a 20× objective lens, and other optical components. Among them, the objective lens reached by the green light of the dichroic mirror converges and illuminates the diamond NV ensemble color center. The resolution of the camera is 1920 × 1080 pixels, where the width of a single pixel is 5.04 μm, and the corresponding pixel size is 0.252 μm. We average the 4 × 4 pixels of the camera to the magnetic field of 1 pixel in this work. The resolution of the magnetic field image is 1.008 µm. The microwave system is mainly composed of a microwave source (N5183B, Keysight, Santa Rosa, CA, USA) and a circulator (D3C2060, DiTom, Fresno, CA, USA). The function of the microwave radiation antenna is to generate resonant effects on the polarized NV color center. The magnetic field system is composed of a permanent magnet, and a precision adjustment structure, which generates the Zeeman split required for imaging and thus responds in resonance with the microwave field at a specific frequency point. The synchronization control system is a control board (PulseBlasterESR-PRO, SpinCore Technologies, Gainesville, FL, USA) that synchronizes the different functions of the microwave source and the camera to achieve continuous ODMR imaging. The image data and the detection result are achieved using a data processing system.

### 2.2. Diamond

The standard geometric length of the diamond sample is 4.5 × 4.5 × 0.5 mm^3^ of 1b-type single crystal diamond from element six. The nitrogen concentration is 100–200 ppm, the lower surface layer presents (100) crystal direction, and the surface roughness is above a submicron. Because 1b-type diamond is doped with a certain amount of nitrogen, it must be processed to complete the production of a relatively high concentration NV color center. The specific operation steps of the processing process are to select high-energy electrons with an energy range of 10 ± 0.5 MeV to irradiate the diamond with an irradiation dose of 9.8 × 10^18^ cm^−2^. The irradiation time is generally 3 h. Then the vacuum annealing was carried out in a constant temperature oven with a temperature of about 850 centigrade, and the annealing time was generally 3 h. Finally, uniformly distributed diamond NV color centers are obtained near the lower surface.

## 3. Simulation Analysis of Radiation Antenna

In this paper, the hollow Ω-type antenna is simulated and designed using HFSS software. The dielectric substrate plate of the hollow Ω-type antenna is Rogers, with a relative dielectric constant of 3.66 and a thickness of 1.524 mm. The hollow Ω-type antenna model and optimized final parameters are given in Figure 1b and Table 1, respectively. The equivalent magnetic field produced by the antenna is uniformly concentrated in the dotted area in Figure 1b. In addition, the feed unit of the hollow Ω-type microstrip antenna adopts a rectangular microstrip transmission line with widths of h1 and h2 to facilitate impedance matching. Figure 1b shows the simulated distribution of the magnetic field in the dashed area of 4.5 × 4.5 mm^2^ at the resonance frequency of the diamond NV color center. According to the transmission line theory, when a single port is input, the other side of the transmission line is an open circuit, and the standing wave will be formed after the superposition of the incident wave and the reflected wave on the transmission line. There are mainly two abrupt changes (bends) in the transmission line structure. The places close to the feed are mainly affected by the feed, while the places close to the open end are not only affected by the front transmission line but also affected by the reflected wave at the open end. Therefore, after the superposition of vector fields, the field near the open end is weaker than that near the feed end. Therefore, the magnetic field distribution is asymmetric in Figure 1b. The straight and radial parts of the antenna form the capacitive and inductive elements, respectively, which determine the resonant frequency of the antenna. The resonance frequency setting depends on related parameters, such as the central hollow radius r1. The range of r1 is set between 4.38 and 4.98 mm, and the corresponding resonance frequency range is between 2.5 and 3 GHz, which is suitable for NV color center resonance testing. Figure 1c shows that the slope is about −2.3 GHz/mm, and the resonance frequency is related.

In order to obtain the regional distribution of the simulated magnetic field, we have extracted the field every 0.2 mm along the two dotted lines in the X and Y-directions in Figure 2a, as shown in Figure 2b. The center of the hollow area is set as the origin of the coordinate, represented by point 1(x = 0, y = 0). The coordinates of the other four points are 2 (x = 0, y = −2.2), 4 (x = −2.2, y = 0), 6 (x = 0, y = 2.2), 8 (x = 2.2, y = 0), and these points to be simulated are in a uniform area. The calculation method of microwave field uniformity is equal to the standard deviation of microwave field intensity divided by the average value of microwave field intensity [29,31]. Clearly, the change of magnetic field intensity is about 5.86% in the area of 4.8 × 4.8 mm^2^, which is calculated with 625 pionts, and the data of 49 pionts on the *x*-axis and *y*-axis in the area are shown in Figure 2b. According to the theory of electromagnetic field and electromagnetic wave, there is a positive linear correlation between the amplitude of the magnetic field and the square root of microwave power.

## 4. Test and Analysis of Radiation Antenna

Using the homemade experimental system in Figure 1a, the hollow Ω-type antenna is characterized by a uniform area. The influence of laser power jitter and NV color center concentration can be eliminated through fixed detection of the laser power, spot size, and the spatial position of the NV color center. Thus, the radiation characteristics at different positions on the surface of the antenna can be accurately tested by changing its relative position. As shown in Figure 2a, nine different single pixel points on the surface of the hollow area are selected and measured, and the corresponding coordinates are 1 (x = 0, y = 0), 2 (x = 0, y = −2.2), 3 (x = −2.2, y = −2.2), 4 (x = −2.2, y = 0), 5 (x = −2.2, y = 2.2), 6 (x = 0, y = 2.2), 7 (x = 2.2, y = 2.2), 8 (x = 2.2, y = 0), and 9 (x = 2.2, y = −2.2). In order to improve the accuracy of the data measurement, each single pixel is measured five times. Figure 3a shows a series of ODMR signals of pixel 1 under different microwave powers. The ODMR spectra in Figure 3a is composed of two resonance peaks because the magnetic field is oriented along the 100 crystallographic direction of the diamond ensemble NV color center so that the magnetic field components projected on the four axes of the diamond NV color center are the same, and the Zeeman splitting amount generated by the four NV axes are the same, which is shown as a pair of peaks in the optical detection magnetic resonance spectrum (a pair of peaks in Figure 4a,b are also formed in this way). The left peak of the ODMR measured at pixel 1 (the black dashed box in Figure 3a) is provided as an example.

The relationship between the FWHM, contrast, and microwave power is shown in Figure 3b,c [32]. In Figure 3b, the fitting formula of FWHM can be written as:(1)$$\Delta v=1/(\pi \cdot T_{2}^{*})+\sqrt{\left(4A_{1}\left(1/T_{2}+\Gamma /2\right)\right)/\left(1/T_{1}+\Gamma +A_{2}\left(\sqrt{A_{3}P}/(1+P/P_{sat})\right)\right)}\cdot \sqrt{A_{3}P}$$
where Δν represents FWHM, P represents microwave power. $T_{2}^{*}$ is the dephasing time; $=1+P_{1}/P_{0},{\;P}_{1}$ is the light power, $P_{0}$ is the optical-pumping saturation power; Γ is the pumping rate, which is determined by the laser; $A_{2}$ is the proportional constant, $A_{3}$ is the ratio of the square of the Rabi frequency to the microwave power; $P_{\mathrm{sat}}$ is the saturated microwave power.

In Figure 3c, the fitting formula of contrast can be written as: (2)$$C=\Gamma \theta /\left(\Gamma +\left(1-\theta \right)/T_{1}\right)\cdot A_{3}P/\left(A_{3}P+\left(1/T_{1}+\Gamma \right)\cdot \left(1/T_{2}+\Gamma /2\right)\right)$$
where C represents contrast ratio, θ is the overall scale constant; $T_{1}$ is the longitudinal relaxation time; $T_{2}$ is the transverse relaxation time. $T_{1}$, $T_{2}$ and $T_{2}^{*}$ are the constants and their values can be known from Refs. [32,33], which are determined by the NV color center, the values are shown in Table 2 below. $A_{1}$, Γ, $A_{2}$, $A_{3}$, θ, $P_{\mathrm{sat}}$ are fitting parameters here and the fitting values are shown in Table 2.

The uniformity of the microwave field is calculated by analogy with the point taking method under the same microwave power in Ref. [31]. Table 3 shows the FWHM, the contrast of nine different points under the same laser power, the external magnetic field of 45 Gs and simulated magnetic field intensity B at nine points. From Figure 3b,c, it can be known that within a certain microwave power range, the microwave power is basically linear with FWHM and contrast. When the microwave power continues to increase, the FWHM and contrast will gradually tend to saturation. Based on the fitting results, 0.126 W of microwave power is selected. Under this power, the microwave power has a linear relationship with FWHM and contrast. With the other conditions the same, the normalized standard deviation of FWHM and contrast at five different positions are 5.61% and 5.71% in the area of 4.4 × 4.4 mm^2^, which are basically consistent with the analysis of the magnetic field strength simulation results, maintaining the uniformity of microwave magnetic field less than 6% in an area of 4.4 × 4.4 mm^2^.

Based on the experimental system in Figure 1a, the CW-ODMR detection principle is used to measure the radiation efficiency of the hollow Ω-type antenna and the straight copper antenna at the same frequency. We measure a pair of ODMR signals in a certain external magnetic field, carrying out five measurements under each microwave power. The measurement positions of the two antenna models are shown in Figure 1b,d, respectively.

Figure 4a,b show the ODMR of the hollow Ω-type antenna and straight copper antenna, respectively. Taking the peak on the left as an example, with a microwave power of 30 dBm, the normalized contrast of the hollow Ω-type antenna is about 4.88%. Here, the straight copper antenna is normalized, and the contrast is about 2.84%. Therefore, the normalized contrast of the hollow Ω-type antenna is 71.8% higher than that of the straight copper antenna, indicating that the radiation efficiency is enhanced.

Figure 4c,d show the comparison of the contrast and FWHM values of the two antennas under different microwave powers. The red dotted line graph is the contrast value and error value of the hollow Ω-type antenna, while the black dotted line graph is the contrast value and error value of the straight copper antenna in Figure 4c. A similar data processing method for FWHM is used in Figure 4d. It can be observed that as the microwave power increases, the contrast and FWHM also increase [34], and the superiority of the Ω-type antenna radiation efficiency is more obvious.

The S11 of the hollow Ω-type antenna and the straight copper antenna is measured using a vector network analyzer (Keysight Technologies N5224A) to determine the bandwidth.

In Figure 5a, the red line and black line represent the measurement and simulation of the hollow Ω-type antenna, respectively. The S11 measurement result of the hollow Ω-type antenna is −40.22 dB at 2.844 GHz, and the simulation result is −47.68 dB near 2.87 GHz. In Figure 5b, the red and black lines represent the actual measurement of the hollow Ω-type antenna and straight copper antenna, respectively. The actual measurement result S11 of the straight copper antenna is −22.8 dB at 2.828 GHz, and the bandwidth is about 83.6 MHz, while the bandwidth of the hollow Ω-type antenna is about 988 MHz. The S11 and bandwidth of the hollow Ω-type antenna are 1.76 times and 11.82 times that of the straight copper antenna, respectively. The results clearly exhibit the effectiveness of the proposed antenna in terms of bandwidth.

## 5. Conclusions

A high-bandwidth, uniform magnetic field, high-radiation antenna for wide-field imaging of diamond NV series color centers was designed in this work. The proposed antenna achieved a 94% uniform area of the magnetic field in the area of 4.4 × 4.4 mm^2^. In addition, compared with the straight copper antenna, under a certain external magnetic field, the normalized contrast of the left peak increased by 71.8%, and the bandwidth of the hollow Ω-type antenna reached 988 MHz when a pair of ODMR signals were measured. All test results demonstrate that the hollow Ω-type antenna effectively complements future quantum sensing antennas and other applications.

## Figures and Tables

**Figure 1 micromachines-13-01007-f001:**
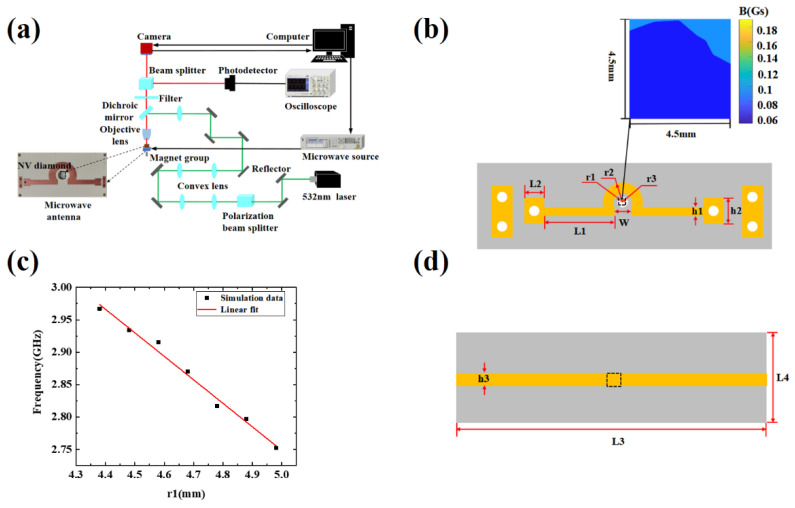
(**a**) Experimental system; (**b**) Schematic diagram of antenna model; (**c**) Relationship between r1 and simulated resonant frequency; (**d**) Straight copper antenna schematic diagram and measurement position (L3 = 59 mm, L4 = 44 mm, h3 = 2 mm, the thickness is 1.524 mm).

**Figure 2 micromachines-13-01007-f002:**
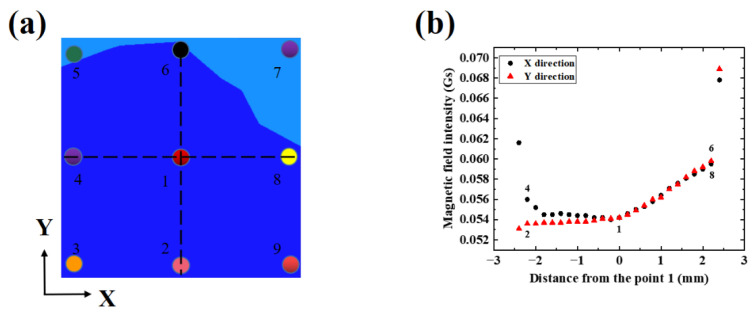
(**a**) Schematic diagram of the hollow area; (**b**) The simulated magnetic field strength in the range of [−2.4 mm, 2.4 mm] of the *x*-axis and *y*-axis, respectively.

**Figure 3 micromachines-13-01007-f003:**
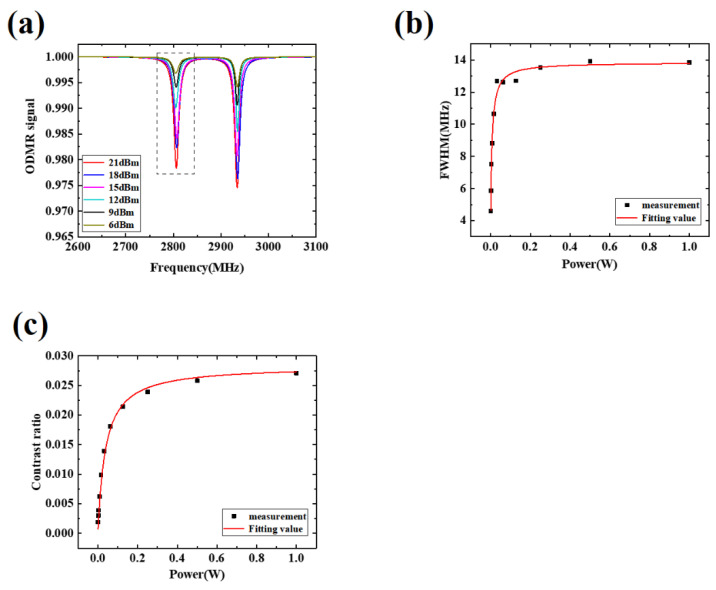
(**a**) ODMR signal with different microwave power; (**b**) Relationship between FWHM and microwave power at pixel 1; (**c**) Relationship between contrast and microwave power at pixel 1.

**Figure 4 micromachines-13-01007-f004:**
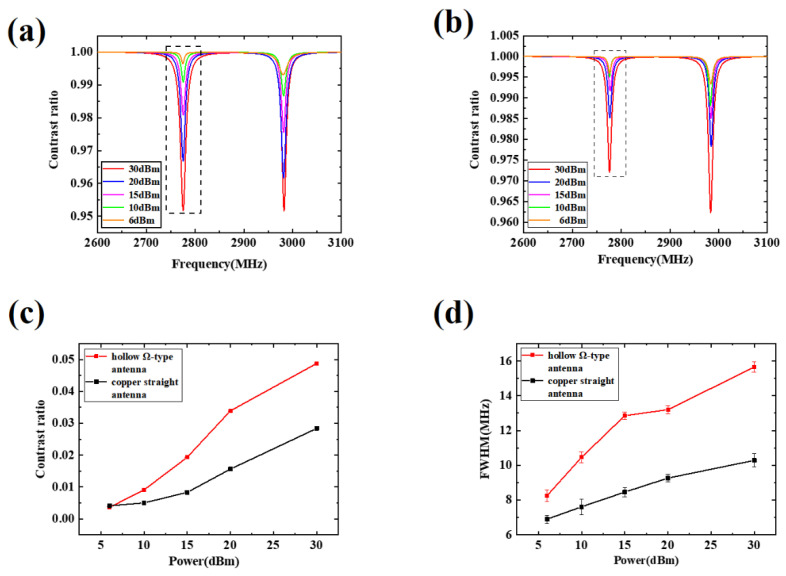
(**a**) ODMR of hollow Ω-type antenna; (**b**) ODMR of the straight copper antenna; (**c**) Contrast value of two antennas under different power; (**d**) Comparison of FWHM value of two antennas under different power.

**Figure 5 micromachines-13-01007-f005:**
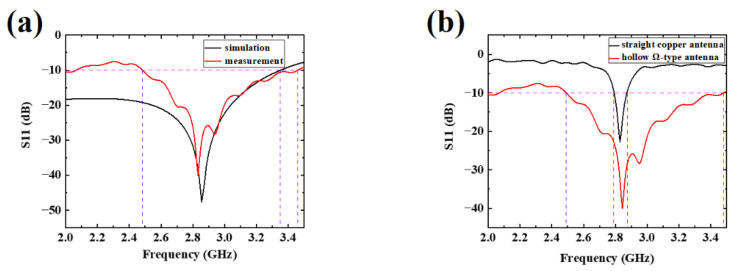
(**a**) Measured and simulated values of S11; (**b**) S11 measured values of two antennas.

**Table 1 micromachines-13-01007-t001:** Antenna parameters (unit: mm).

Parameters	r1	r2	r3	W	h1	L1	h2	L2
Value(mm)	4.68	9.24	3.22	6.62	3.22	18	5	4.3

**Table 2 micromachines-13-01007-t002:** The constants and fitting parameters in Equations (1) and (2).

Constant	Value	Fitting Parameter	Value
$T_{1}$	~1 ms	$A_{1}$	50 (8)
$T_{2}$	~1 μs	Γ	$0.54\;(9)\;{\mu s}^{-1}$
$T_{2}^{*}$	~0.1 μs	$A_{2}$	1.6 (2)
		$A_{3}$	$18\;(5)\;{\mathrm{MHz}}^{2}{\mathrm{mW}}^{-1}$
		θ	0.0284 (6)
		$P_{\mathrm{sat}}$	198 (27) mW

**Table 3 micromachines-13-01007-t003:** Statistical analysis of FWHM, contrast, and simulated magnetic field intensity B at different points in Figure 2a.

**Point**	**1**	**2**	**3**	**4**	**5**	**6**
FWHM (MHz)	12.026	13.206	12.656	12.424	13.988	12.098
contrast	0.0248	0.0228	0.0232	0.0248	0.0254	0.0258
B (Gs)	0.0542	0.0536	0.0556	0.056	0.061	0.0598
**Point**	**7**	**8**	**9**	**Mean**	**Standard** **Deviation**	**STD/** **Mean** **(%)**
FWHM (MHz)	13.689	13.658	12.658	12.9337	0.726	5.61
contrast	0.0266	0.0226	0.0246	0.0245	0.0014	5.71
B (Gs)	0.0623	0.0595	0.0565	0.0576	0.0031	5.38

## Data Availability

Not applicable.

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
