# Peer review of "Design of a High-Bandwidth Uniform Radiation Antenna for Wide-Field Imaging with Ensemble NV Color Centers in Diamond"

_micromachines, 2022, doi:10.3390/mi13071007_

Round 1

Reviewer 1 Report

The article is focused on the development of a antenna-like structure for generating oscillating magnetic field at microwave frequencies suitable for  the wide-field sensing with nitrogen-vacancy (NV) color centers in diamond. The main accomplishment is the uniformity of the generated field over a large area of (several mm)^2. While this feature may be valuable for many NV experiments, the Authors need to address several other important factors, listed below, before this work may be published.

1. Authors should clarify that for the NV experiments it is the oscillating magnetic field component of the EM field that is of importance. Even though many works use the term 'MW antenna', it is the near-field that is used, not the radiated (faar-field) part of the EM field. 

2. Large-area (and large-volume) boradband & uniform MW structure has been demonstrated in:

Broadband loop gap resonator for nitrogen vacancy centers in diamond, Review of Scientific Instruments 89, 094705 (2018); https://doi.org/10.1063/1.5037465

3. Omega-type antennas are widely used in the NV community and appropriate references to previous works should be provided. Some can be found in the reference provided above. 

4. In general, making the radius of the loop part larger results in more uniform field in the center, however, at the cost of lower field amplitude. Therefore, the figure-of-merit is the microwave B-field per watt of MW power. Alternatively, one can provide the Rabi frequency instead of B-field as they are proportional. Authors should compare the efficiency of their structure with the relevant published ones in this aspect.

5. Details of both omega and straight-line antenna geometries need to be provided clearly. The enclosed image in Fig.1 is of poor resolution and simply does not allow finding the parameters for the omega structure geometry. Each parameter needs to be clearly visible on the plot and/or named in text or caption. All dimensions for the straight line need to be also provided clearly.

6. What is the substrate used for the boards? (type, thickness, epsilon parameter needed)

7. The ODMR spectra in Figs3&4 consist of 2 resonances instead of 4 pairs. Is that because the static B-field oriented along the 100 crystallographic direction of the diamond plate? Or the sample has been prepared to have NVs preferentially aligned along one crystallographic orientation? Please explain. 

8. The oscillating B-field produced by the omega structure has the direction normal to the board surface. On the other hand, for the straight line, the normal component should disappear. That may lead to completely different contrasts observed in the ODMR spectra even at the same B-field amplitude! That depends on the projection of the oscillating magnetic field vector on the NV axes in the diamond, so is also related to my question #7. 

Therefore, the statement "Compared with a copper straight antenna, the radiation efficiency of the proposed antenna is 71.8% higher" is not fully supported.  

9. Refs. 18 and 21 do not seem to present examples of NV-diamond based sensing. Please check these. 

10. I believe the statement 'According to the theory of electromagnetic field and electromagnetic wave, there is a linear positive correlation between magnetic field intensity and microwave power. Therefore, microwave power is equivalent to magnetic field intensity." is incorrect. The B-field amplitude (as well as Rabi frequency in NVs) should be proportional to the square root of the MW power.

11. Finally, some English grammar editing is required. 

Reviewer 2 Report

The authors present a wideband microwave hollow antenna that enables one to perform ODMR on an ensemble of NV centers efficiently and uniformly.  I have a couple of comments and questions below. 

The figures are too faggy.  In particular, one could not recognize the parameter symbols in Fig1 (b). 

I could not understand how the hollow antenna couples to the external coaxial line(s).  The reader would appreciate it if the authors also presented a photograph of the whole device, i.e., the PCB with a microwave connector. 

Why is the simulation curve in Fig. 5 (a) asymmetric?  I guess it is because of the open end as described on P3.  Why not terminate the open end by 50 ohms if that is the case? 

To be fair, please mention that this antenna device has a 10 dB bigger insertion (internal) loss than the straight antenna (Fig. 5b), which I guess is the price to pay for the bandwidth. 

Reviewer 3 Report

In this work, Li et al. introduce the results obtained, in terms of radiation efficiency and bandwidth of an antenna specifically designed for wide field imaging based on NV diamond color centers.

In the last few years, the study of NV centers in diamond and of their possible applications has been gathering much attention from the scientific community. It's definitely a hot topic. 

Authors contribute to this research activity by demonstrating that a hollow Omega-type antenna surrounding diamond is more efficient than a straight one, ensuring at the same time a wider bandwidth. Demonstration is rigourously based on experimental results and on their matching with simulations. 

I think that the paper deserves to be published in Micromachines after revision. 

In my opinion, there is one important aspect which has not been properly taken into account by the authors. I mean, there is no description of the diamond used in their work.  All the details regarding the diamond sample properties are missing, and should be mandatorily added in the revised version of the manuscript: thickness, lateral dimensions, grade (mechanical, optical, electronic?), orientation. Also, is it a single-crystal or a polycrystal, or a nanodiamond? How NV centers have been created? Are those present in the as-grown sample or have they been created by ion implantation or femtosecond laser irradiation? All these info should be added in my opinion, for consistency with other works present in the literature on the same topic.

Round 2

Reviewer 1 Report

I am happy with the corrections made. Though I'm still missing the information on the direction of the applied bias (dc) magnetic field in the text. It is present in the Authors reply (#7) but I couldn't find this reflected in the text. 

Figure quality in the PDF I received is still poor. 
